# Optimal Period for Winter Mountain Tourism in Romania

**Ciprian Ioan Rujescu** 

Department of Management and Rural Development, Faculty of Management and Rural Tourism, Banat's University of Agricultural Sciences and Veterinary Medicine "King Michael I of Romania" from Timisoara, 300645 Timisoara, Romania; rujescu@usab-tm.ro

**Abstract:** Observations in Romania show that the months of January and February are starting to show an increasing interest for tourists in areas known for winter sports involving snow. This observation is at odds with the period hitherto considered traditional for winter tourism in Romania, i.e., from the end of December to the first few days of January, when school holidays and employee holidays are frequently scheduled. Analysis of the climatic data of recent years shows a shortening of the period when natural weather conditions are favorable for this type of tourism. In this paper it was statistically determined that the maximum share of negative temperature coverage of Romania's territory tends to occur at present in the second half of January. It is therefore necessary to correlate the school and labor law timetables with the new climatic conditions and other measures to adapt to current conditions.

**Keywords:** winter tourism; snow; climatic change; temperature; winter sports

## 1. Introduction

Climate change is still an insufficiently understood topic. Given the empirical observations that society frequently and often unintentionally assimilates, this subject is a current scientific concern. The unpredictability of change leads to a real difficulty in fitting evolution into a mathematical pattern. This often gives rise to opinions that lack credibility. Scientific opinions are sometimes contradictory and even if they clearly indicate a slight increase in average temperatures in some areas, they do not provide unanimously recognized proof and, even more, they cannot prove whether this increase in average temperatures is transient or will be of a long-term nature.

Today, however, tourism cannot ignore the implications of this issue. There are segments of tourism that are differently sensitive to climate change. For example, changes in average temperatures over the course of a tourist season, increasing by even one degree, do not immediately affect the perception of tourists during the summer at the seaside. In coastal resorts where there are buildings covering specific areas, climate changes do not lead to changes immediately felt by tourists. Furthermore, the winter season in high altitude mountain areas, where low temperatures prevail, are not strongly affected by the low temperature rise.

There is, however, a segment of mountain tourism, namely that practiced at low altitude, sometimes even at the relief limit between hill and mountain, which requires the existence of snow and which is much more sensitive to temperature increases. At this level, the seemingly insignificant difference in temperature of just one degree is often the difference between frost and thaw, rain and snow, an icy access road or an easy one, and a shorter or longer winter tourist season. The list of components affected by these changes is long, and it is understandable that both tourists and tourism entrepreneurs are directly affected.

The study is conducted for the territory of Romania. The country is geographically positioned in the continental European area and has a relief with similar characteristics to other states. This allows the results to be extended to similar situations.

Currently in Romania there are few locations at altitudes above 2000 m that have been developed for the purpose of winter tourism, more precisely for sports involving snow. Most are found at altitudes between 700–1800 m, sometimes even lower. This altitude segment is representative for winter mountain tourism in Romania. Concerning the condition and efficiency of ski slopes in Romania today, due to the decrease in the period of time the ground is covered with snow, it was determined that only about 18–20 percent were open for a period of 100 days during two winter seasons, 2016–2017 and 2017–2018, and this induces a low economic profitability [1].

Climate change is directly affecting this type of tourism in multiple locations worldwide [2]. Melting glaciers and warming mountains are leading to significant changes in the lives of some local communities. The literature indicates implications for their cultural identity [3]. A number of vulnerabilities such as avalanches, floods, and landslides in mountain tourist areas may be due to the climate. These factors can indirectly contribute to increasing insecurity and to the depreciation of the tourism industry [4]. Moreover, it also has indirect implications such as a decrease in the physical activity of the population, even correlating with human health, and difficulties in organising sports competitions [5,6].

A common solution to compensate the lack of natural snow is the classic use of artificial snow. However, the efficiency of snowmaking is difficult to characterize in a unified way and there are currently multiple concerns for its optimization [7–9] and multiple contradictions as well. Snowmaking could be a risk factor for the environment due to high water consumption, which has been analyzed in mountain resorts in France and Switzerland. The use of energy, water and fossil fuels can compete with the basic needs of local communities and could have contrary effects. The presence of microorganisms can be found in artificial snow as a result of insertion through transport pipes or sprayed air [10–13]. An analysis of how tourist areas in the Eastern Alps are coping with climate change reveals that different locations are affected differently, but the current low snow cover induces the need for snow production [14]. In Turkey, it was also found that currently only high-altitude resorts are not affected in terms of snow cover. Even with the various drawbacks, snowmaking is a solution that compensates for the natural deficit in low altitude areas [15]. However, the strategies must be adapted according to local particularities [16].

However, the production and use of artificial snow cannot be economically viable when the ambient temperature is positive. The outdoor environment temperature is an uncontrollable factor for winter resort managers, whereas snow has become a controllable factor under the right temperature conditions.

The aim of the study is to assess the extent to which climate change has changed the calendar period in which snow sports can be practiced in Romania.

## 2. Methodology

The analysis is based on statistical data provided by the National Institute of Statistics Romania (NISR), relating to overnight stays in tourist reception structures by regions and counties [17]. Data on average monthly temperatures reported by the National Meteorological Administration Romania (NMAR) were processed [18]. NMAR presents in the form of heatmaps details regarding the coverage of the Romanian territory, with the average monthly temperatures. The GIMP 2.10.30 application equipped with the Histogram Dialog function can estimate the statistical color distribution of the selected layers [19]. This procedure allows you to approximate the weight of surfaces in an image. Specifically, a ratio was made between the area marked with the same color and the total area.

The satellite images from Sentinel Playground Hub were processed for comparative purposes [20].

Statistical calculations were performed using SAS Studio software [21] or Past 4.03 [22].

The observations show that between the four months of November–January that cover the winter period, there are some differences regarding the number of overnight stays in tourist structures. The hypothesis that significant differences existence, namely in the fact that tourists are not evenly distributed over the four months, was the subject of statistical

analysis. For the purpose of testing the differences, the Kruskal–Wallis test was used, applying the procedure SAS Studio 3.8, non-parametric one-way ANOVA.

Figure 1 shows the research procedure.

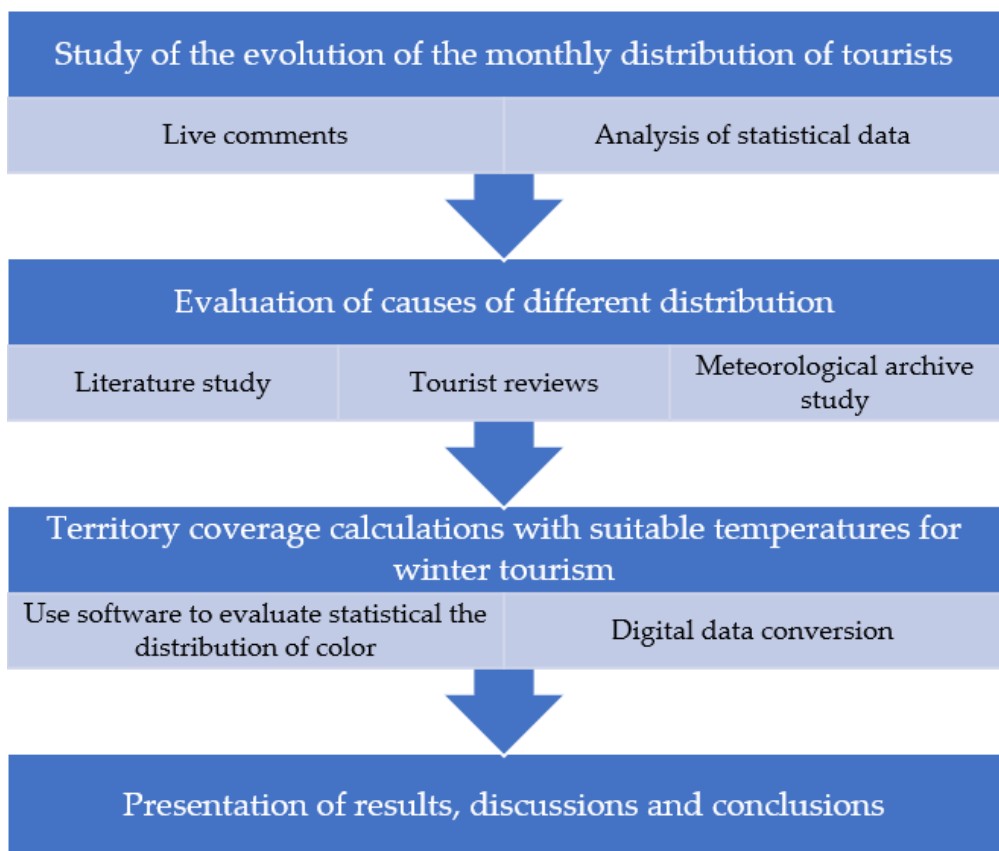

**Figure 1.** Presentation of the research procedure. Source: Own representation.

The surface area of Romania is 238,398 square kilometres. The share of hilly and mountainous relief cumulatively exceeds 65 percent, of which mountainous relief is over 30 percent [23]. An image of the Carpathian Mountains in Romania is shown in Figure 2. Their length is approximately 900 km. Three major groups of these mountains are present in this area: the Western, Southern and Eastern Carpathians. The highest peak is Moldoveanu located in the Făgăraș Mountains of the Southern Carpathians group, with a maximum altitude of 2544 m. In the Eastern Carpathians, the highest peak is Pietrosu, reaching a maximum height of 2303 m. In the Western Carpathians, the highest peak is Curcubăta Mare, 1849 m [24].

The case study focuses in particular on a geographical region considered representative for this type of tourism, Caras Severin County. Here there are two important ski areas. The first one is located in the Semenic-Crivaia area and starts at an altitude of about 740 m in the vicinity of Văliug, respectively at about 1400 m for the ski slopes in the Semenic Mountains. Another ski area is located in the Muntele Mic area, at an altitude of over 1500 m. The maximum altitude found here is 1802 m. The mentioned ski areas are the most important winter tourist attractions in this county, being the locations that attract the largest number of tourists in Caras Severin County.

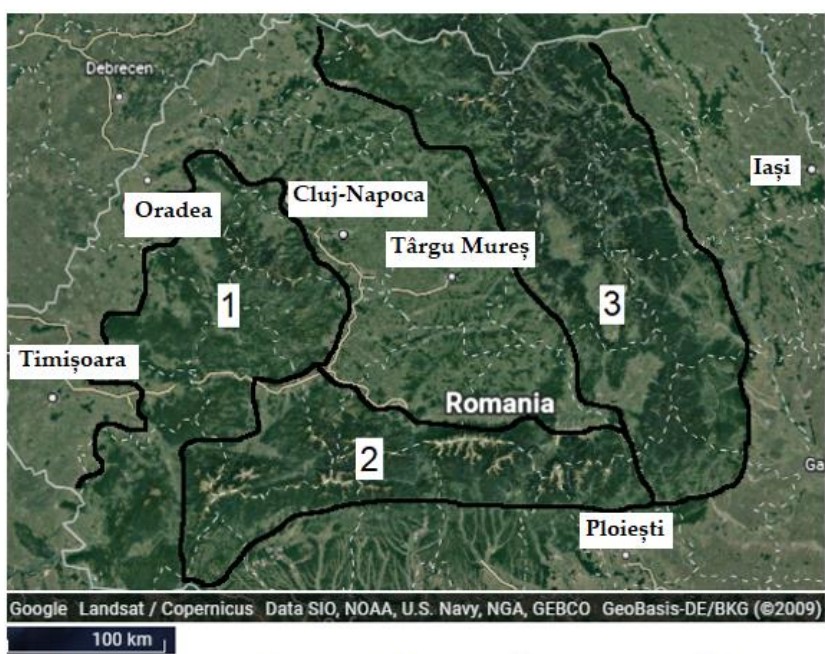

(1) Western , (2) Southern and (3) Eastern - Romanian Carpathians

**Figure 2.** Carpathian Mountains in Romania. Source: Own processing based on Google Earth [25] image.

## 3. Results

A large part of the mountain locations for winter tourism in Romania are directly affected by the reduction of snow cover, as they are located at lower altitudes. At altitudes above 2000 m the snow cover is frequently deposited from October onwards and often continues uninterrupted until April, sometimes even extending beyond this period. While in summer access to areas at this altitude is facilitated by the existence of cable-supported transport or even minimal road infrastructure, winter access is difficult and often fraught with risk. Moreover, accommodation at this altitude has traditionally been designed for use and operation in the summer season.

In fact, a few years ago, being located at low altitude brought many economic advantages. The short access time from neighbouring towns, the road infrastructure which does not require very large investments for access at low altitudes, and the utility networks for electricity, water, sewage and even telephone supply until recently gave these locations an economic advantage based on reduced costs. The winter tourist season often lasts from November and sometimes even until early April. Experience in recent years suggests that the season is shortened to mid-January and the first part of February, corresponding to the period when the ground is covered with snow at altitudes of 700–1800 m. The months of November, December or, on the other hand, the end of February or March are characterized by a real uncertainty as to the possibility of natural or artificial snow cover.

This is also immediately reflected in the distribution of tourists during the winter months. The analyzed NISR statistical data [17] refer to the number of overnight stays in tourist accommodation facilities in the entire Caras Severin County. This indicator expresses the number of nights for which a tourist is registered in a unit that offers accommodation. The important share of tourists in the two ski areas in this county allows approximating some distribution trends in the winter months. November is the month in which the number of overnight stays predominates, followed at a considerable distance by tourists from December to February. November is the month in which the first snows appear sporadically and is a long-awaited moment for winter sports enthusiasts. During this period a large number of tourists come to mountain resorts even though the snow cover is not consistent and temperatures are variable. Figure 3 shows the distribution of the number of overnight stays in tourist accommodation facilities in Caras Severin County during the winter months for the whole period 2010–2021.

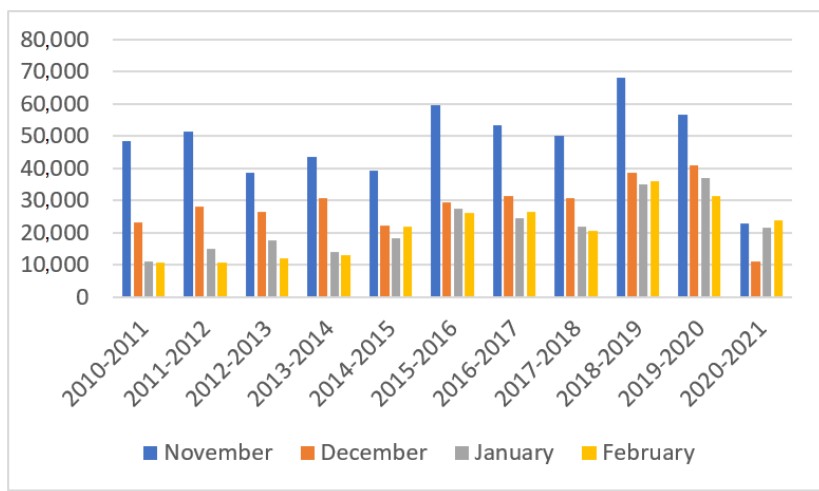

**Figure 3.** Distribution of the number of overnight stays for tourist accommodation facilities during winter 2010–2021 in Caras Severin County, Romania. Source: Own graphical representation based on NIS statistical data.

The distribution by winter months of these statistics differed significantly. The distribution of the calculated Wilcoxon score expressed as ranksums of the number of overnight stays of each month from November to February for the cumulative statistical data from 2010–2021 are shown in Figure 4. Over the entire time span of the last 10 years, the number of overnight stays in November is often the highest. The December values for the number of overnight stays are also often higher than in January and February. The null hypothesis that the distributions are unchanged over the four months is rejected, $\chi 2 = 22.39$ with $p < 0.001$. So, over the studied interval the monthly data are different.

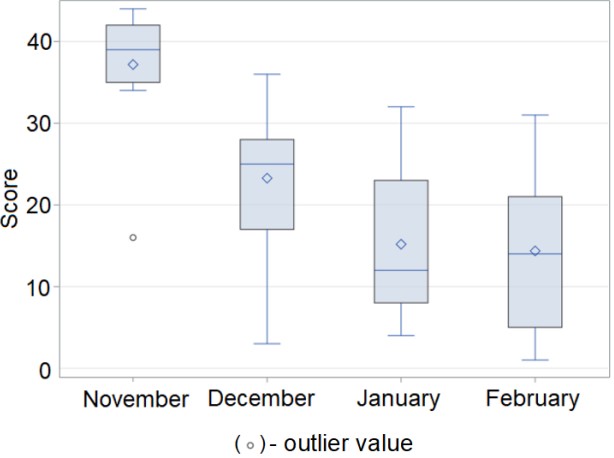

( ∘ )- outlier value

**Figure 4.** Distribution of Wilcoxon scores (ranksums) of the number of overnight stays in tourist accommodation facilities during winter 2010–2021 in Caras Severin County, Romania. Source: Own graphical representation based on NIS statistical data using SAS Studio.

However, looking at the trend in the evolution of the data distributed by months from the winter of 2010–2011, 10 years ago to the present, we see a change in the sense of a rapid increase in the number of overnight stays in the months of January and February on a rather constant background of this indicator in the months of November and December. Figure 5 shows the trends in the number of overnight stays in linear form. An important change is observed in December. Ten years ago, the number of overnight stays in tourist accommodation facilities in Caras Severin County in December was more than double the number of overnight stays in January or February of the same winter. The trend in the number of overnight stays in December shows a slight increase, even negligible.

The slope value of the regression line for December is 313.95, indicating a year-on-year increase of about 10 overnight stays per day. However, the upward trend is much faster for January (slope 1951.2, about 62 more overnight stays per day than the previous year) and February (slope 2179.3, about 77 more overnight stays per day than the previous year). Thus, at present, according to these trends, the number of overnight stays for December has ended up being lower than in January and also lower than in February. The situation is seemingly paradoxical because December is considered a symbol of winter, with holidays accompanied by non-working days and school holidays.

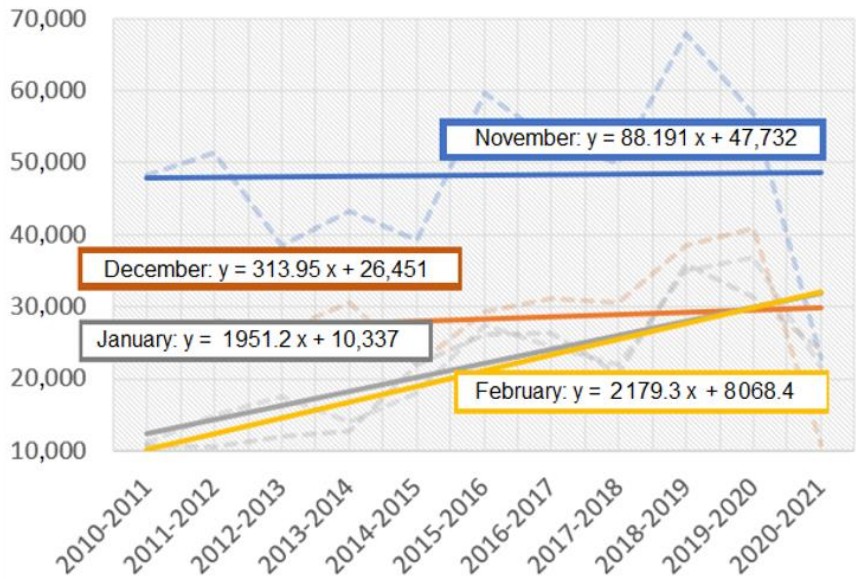

**Figure 5.** Trend in the number of overnight stays for tourist accommodation facilities during winter 2010–2021 in Caras Severin County, Romania. Source: Own graphical representation based on NISR statistical data.

One factor explaining the changes in the distribution of tourist overnight stays in the winter months is precisely the shortening of the snow season to January and February.

Currently, tourists do not schedule trips under uncertainty. The large number of websites specialized in monitoring ski slopes, snow cover, increasingly accurate weather forecasts, and real-time reviews of other people, make the indicator overnight stays a suitable descriptor of reality. Most often in the latter period, snow has settled at this altitude from the first days of January and continues throughout February. During November and December, the ground is only casually covered with snow. For comparison purposes, the Sentinel Playground Hub satellite images, infrared based on bands 8,4,3, Sentinel-2, for the Semenic-Crivaia area are shown as a pair for the months of December and February (Figure 6). They show lower temperatures in early February compared to early December.

Infrared detection is a useful practice in determining terrestrial temperatures. The possibility of accessing the imagery history allows a deeper understanding of local weather phenomena [26,27].

The heatmap images provided by NMAR under Climate Monitoring, Monthly Mean Temperatures, are arranged in parallel for comparative purposes over the winter months in different years in Figure 7. Using the GIMP 2.10.30 application, the weightings of areas in Romania with negative monthly mean temperatures, i.e., below $-4\,°C$, were extracted. The aim is to characterize the winter months in terms of average temperatures and to detect in which months Romania's territory has the largest areas with negative temperatures.

Many winter tourist resorts in Romania have ski slopes equipped with artificial snowmaking facilities, but their use becomes very expensive or even impossible for a large part of the winter due to the low temperatures.

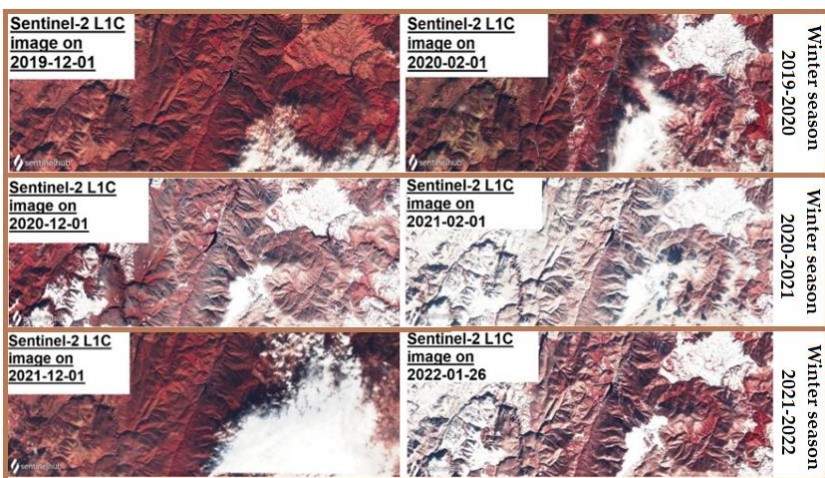

**Figure 6.** Satellite images of the Semenic-Crivaia area taken in December and January/February of the same season. Source: Own representation by processing of Sentinel Playground Hub images.

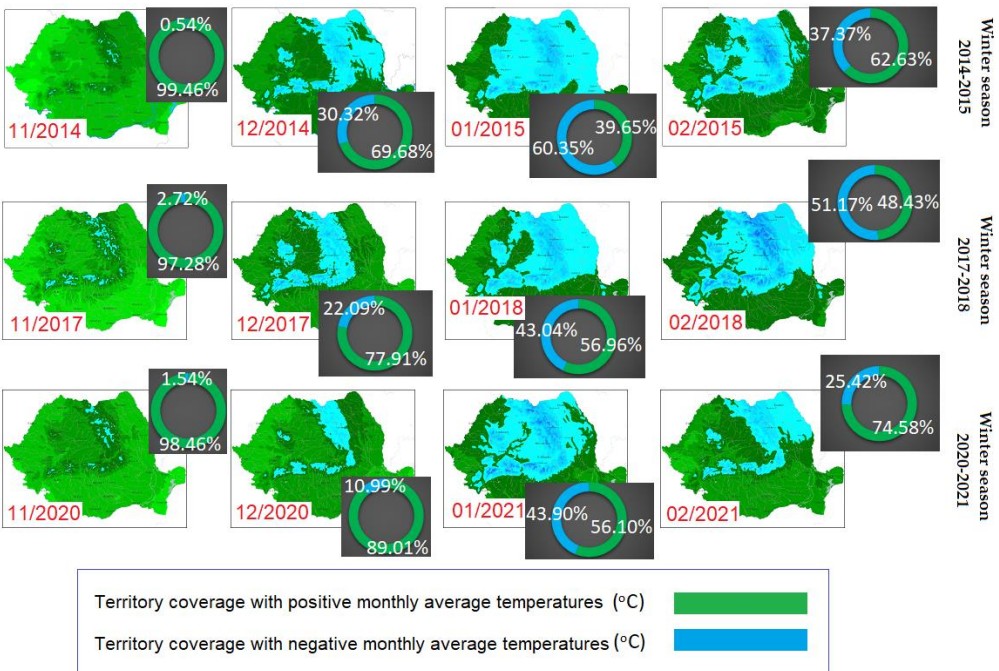

**Figure 7.** Share of Romania's area with negative temperatures in November–February. Source: Own calculation by processing of NAMR images using GIMP 2.10.30 (climate monitoring).

The values corresponding to the cold months of the year, from November to February, for the winter period of 2014–2015 and until 2020–2021 are described by the statistical summary in Table 1. It should be noted that in November, areas with positive average temperatures predominate and in isolated cases, negative average temperatures occur in areas with high altitudes. For the seven winter seasons, the average share of Romania's area covered with negative temperatures in November is 1.93 percent. Only about 0.02 percent of the country's surface was covered by temperatures below −4 °C during this period. The month of December is characterized by negative average temperatures in the Carpathian arc area, most often in the Southern Carpathians and the Western Carpathians in the highlands. During the same time interval, about 34 percent of the country's area had negative temperatures in December. Only about 4.67 percent of Romania's surface was covered with average temperatures of less than −4 °C in December. In January, negative average temperatures also occur in the Eastern Carpathians and even in the vicinity of

the Carpathian arc, thus also at lower altitudes. January is characterized by the highest proportion of the country's surface covered with negative temperatures (70.33 percent). More than 21.98 percent of the area had average temperatures below −4 °C in January. February also, monitored over the seven seasons, shows areas totalling a high proportion of the country's surface with negative temperatures (27.34 percent). More than 5.6 percent of Romania's surface area had temperatures below −4 °C in February.

**Table 1.** Statistical summary on the percentage of territory coverage with negative average monthly temperatures during winter, in the period 2014–2021 in Romania.

| Analysis Variable: Negative Temperature Territory Coverage (Percent) | | | | | | |
|---|---|---|---|---|---|---|
| Month | Type | Mean | Std. Dev. | Minimum | Maximum | Median |
| November | $t < 0$ | 1.93 | 2.62 | 0.11 | 7.52 | 0.95 |
| | $t < -4$ | 0.02 | 0.04 | 0.00 | 0.10 | 0.00 |
| December | $t < 0$ | 34.54 | 29.73 | 10.99 | 80.95 | 22.08 |
| | $t < -4$ | 4.67 | 6.70 | 0.19 | 20.28 | 2.14 |
| January | $t < 0$ | 70.33 | 25.95 | 43.03 | 100.00 | 60.35 |
| | $t < -4$ | 21.98 | 28.89 | 5.45 | 91.72 | 9.57 |
| February | $t < 0$ | 27.34 | 15.87 | 3.93 | 51.16 | 25.42 |
| | $t < -4$ | 5.36 | 4.47 | 0.14 | 13.04 | 3.56 |

Source: Own statistical processing using SAS Studio of indicators calculated for NAMR images by GIMP 2.10.30 (climate monitoring).

A direct comparison between the minimum, maximum, median, and mean values of the weight of areas with negative temperatures in Romania can be seen in Figure 8 containing boxplots.

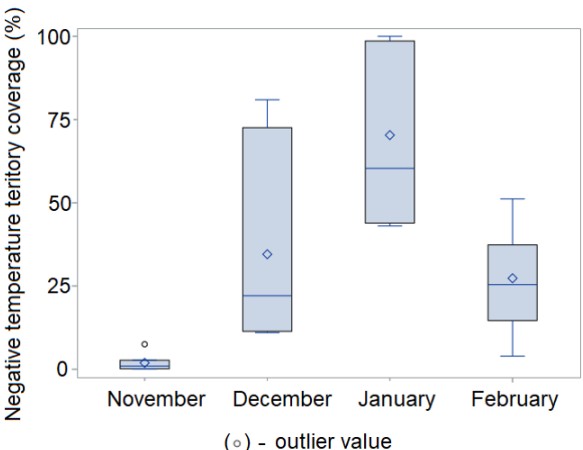

**Figure 8.** Boxplots of the percentage of land cover with negative mean monthly temperatures during winter in the period 2014–2021 in Romania. Source: Own graphical representation based on NAMR statistical data using SAS Studio.

The following function was obtained by using the mean values contained in Table 1:

$$f(d) = -0.0002159 \, d^3 + 0.02458 \, d^2 + 0.2317 \, d - 4.679 \qquad (1)$$

The notation d represents the chronological number of days in the period 1 November–28 February and f(d) the function indicating the degree of coverage of Romania with negative temperatures on a given day in the mentioned interval. The graphical representation is shown in Figure 9. Starting from the expression of the function, the value leading to its maximum, d = 80, was immediately determined. Thus, it is expected that the maximum area of the country where negative temperatures are recorded occurs in the vicinity of 19 January.

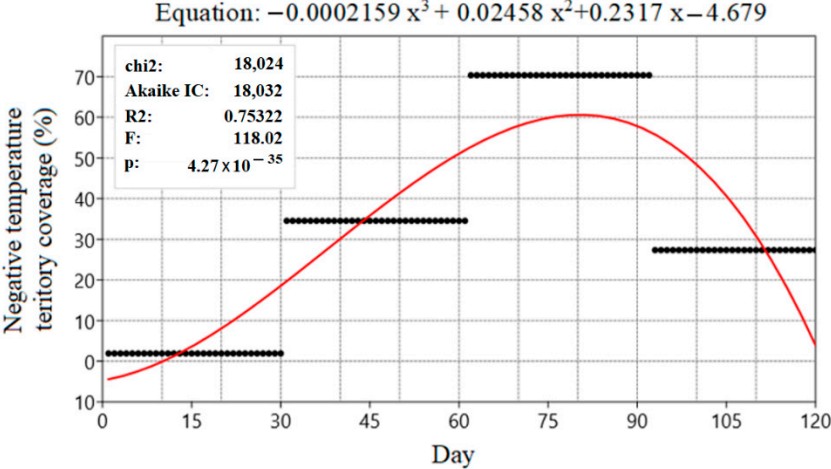

**Figure 9.** Monthly evolution of the average percentage of territory coverage with negative average monthly temperatures during winter in Romania in the period 2014–2021. Source: Own graphical representation using Past 4.03 of indicators calculated for NAMR images by using GIMP 2.10.30 (climate monitoring).

The mathematical model thus determined was the subject of the analysis based on the simulation of the estimated land cover of Romania with negative temperatures. The data are partially presented in Table 2. As stated before, it should be noted that on 1 December, only about 19 percent of the surface of Romania is covered by negative temperatures. On 24 December, when the winter holidays start, the estimated negative temperature coverage is about 45 percent. The end of February brings a low percentage of negative temperature coverage of less than 4 percent of the country's surface. The estimated period of at least 60 percent coverage is between the 15 and 23 January. It is therefore assumed that this is the optimum time for winter sports requiring snow.

**Table 2.** Statistical simulation of the proportion of Romania's territory covered by negative temperatures.

| Day | Date | Estimated Coverage (Percent) | Residual |
|---|---|---|---|
| 1 | 01-November | - | - |
| 31 | 01-December | 19.69 | 14.86 |
| 54 | 24-December | 45.51 | −10.96 |
| 59 | 29-December | 50.21 | −15.66 |
| 75 | 14-January | 59.86 | 10.47 |
| 76 | 15-January | 60.12 | 10.22 |
| 79 | 18-January | 60.57 | 9.77 |
| 80 | 19-January | 60.61 | 9.73 |
| 81 | 20-January | 60.60 | 9.74 |
| 83 | 22-January | 60.42 | 9.92 |
| 84 | 23-January | 60.24 | 10.10 |
| 98 | 06-February | 50.86 | −23.52 |
| 100 | 08-February | 48.36 | −21.02 |
| 119 | 27-February | 7.10 | 20.25 |
| 120 | 28-February | 3.96 | 23.39 |

Source: Own statistical processing using Past 4.03 of indicators calculated for NAMR images by using GIMP 2.10.30 (climate monitoring).

The absence of snow induces immediate reactions in the offer of tourist services in Romania. For example, a large number of slopes have been equipped with snow production facilities. The service use has also been extended by equipping it with light installations that allow use during the dark period of the day. The Semenic-Crivaia ski area is currently undergoing a major modernization campaign. The slope in the upper area of Semenic,

from an altitude of 1400 m, has already been unified with the one located at the base of the domain, at Crivaia at about 740 m. Its length thus reaches 6000 m by unification. This approach is a substitute for the absence of snow at the lower base of the slope.

## 4. Discussion

It was observed that both the demand and the supply in the field of tourist services depend on the environmental conditions but also on the management of the different situations [28]. Adaptation solutions are context-dependent and the adaptability of some regions is different [29].

In Romania, school holidays and a large part of the holidays or days off for employees start around Christmas, at the end of December. The second Thursday of January is the period when school holidays and, to a large extent, employees' holidays end in Romania. This period seems to have been less favorable for winter tourism which requires snow in recent years. A great deal of information reported in the national press and on websites specialising in monitoring the state of ski slopes and opinions of people interested in winter sports indicate that the period when snow is present in Romania is out of sync with the holiday and vacation period.

The school summer vacation in Romania takes place approximately between 15 June and 15 September. It is a much longer holiday compared to other European countries. The holiday period in the first two weeks of September could be moved to January. If the winter holidays now rarely exceed 15 January, the two weeks in addition would cover the entire month of January. Similar changes could be applied to the calendar of economic activities.

Even if the events of climate change are considered uncertain, analyzing ex ante, holiday relocations would not cause significant disruption to the education or economic system. The Christian holiday season is also covered by the holiday season. The period 1–15 September is less requested for summer tourism in Romania. The index of net use of accommodation capacities in Constanța County, located on the Black Sea, a highly sought-after summer vacation destination, decreases from about 68 percent in August to 37 percent in September. These values were calculated as the average for the years 2017–2021 [30]. The decrease in the occupancy rate in September occurs even when, from 1 September, in Romania the program "Seaside for all" occurs, characterized by reductions in accommodation rates with values sometimes of 70 percent [31]. According to the Eurydice Report on the organization of school time in Europe [32], for the school year 2021/2022, out of the 37 countries/regions analyzed, in 10 countries the school year starts in August and in 13 countries on 1 September. In the countries of Southern Europe, but also in Luxembourg and Romania, the school year begins in the second half of September.

There are currently some legislative initiatives on linking the structure of the school year to climatic conditions.

The idea of adapting to climate change is a frequent topic in the literature. A recent analysis of the impact of climate change on tourism in the Black Forest shows that rising temperatures here have not significantly affected tourism demand or management decisions. However, this tourist area offers multiple activities outside of winter sports [33]. Winter resorts located in Austria at low altitudes lose out in competition with others at higher altitudes. Compensation could occur by diversifying tourism offers other than those involving natural snow [34]. Adaptation or non-adaptation of some local areas will be the factors that make or break the success or failure of winter tourism and not all of the industry is affected [35]. In Romania, at present, there are a small but growing number of entrepreneurs in the tourism industry who have started major investments at altitudes above 1500 m in health, spa, or wellness facilities to provide a substitute for winter sports.

## 5. Conclusions

The southernmost point of Romania has a latitude of 43°37′07″ N, the northernmost 48°15′06″ N, and the highest point is 2544 m. The climate is continental with both polar and sub-tropical influences [24]. The low altitude of the mountain resorts but also the geo-

graphical positioning makes it vulnerable to climate change, even of a small value. Today's changing climate makes it necessary to diversify the services that tourism entrepreneurs offer to low-altitude mountain locations. This is because tourism services involving snow or negative temperatures are characterized by a high degree of uncertainty.

The structure of the school year in Romania, i.e., the calendar period of the current winter holidays, does not allow winter sports to take place except by chance or only in high altitude locations. The apparent shortening of the duration of negative temperatures only coincidentally overlaps with the Romanian winter holiday period. There are few places at altitudes above 2000 m, so linking the traditional period of holidays and winter holidays to the climatic reality of the moment becomes a useful and easily applicable solution. Changing the calendar of school holidays in the sense of extending the winter holidays, and respectively reducing the summer holidays, leads to a structure of the school year close to European countries with similar geographical characteristics.

The solutions for weekend winter tourism are quite difficult to implement due to the geographical location, as some large cities are located at a considerable distance from the high mountain area. The congestion caused by an unprepared mountain road infrastructure often discourages many people from choosing mountain destinations in Romania in winter. In order to increase the frequency of tourists in the winter months in the conditions of shortening the duration of the season, in addition we recommend the following. An applicable solution may be the development of public transport routes with a special program that will allow tourists access to mountain locations without using their own car. Another solution is to create the legislative framework for the organization by the educational institutions of some tourist trips at times when the meteorological conditions are favorable for the practice of winter sports. This can be done even outside of holidays with the possibility of recovering courses.

This study presents certain limitations regarding the changes in the demand and supply of tourism during the COVID-19 pandemic in the period 2020–2021. Another limitation is due to the accuracy of the estimation of the territory coverage with fixed temperatures.

**Funding:** This research received no external funding.

**Data Availability Statement:** The present study did not report any data.

**Acknowledgments:** This research paper is supported by the project "Increasing the impact of excellence research on the capacity for innovation and technology transfer within USAMVB Timișoara" code 6PFE, submitted in the competition Program 1—Development of the national system of research—development, Subprogram 1.2—Institutional performance, Institutional development projects—Development projects of excellence in RDI.

**Conflicts of Interest:** The author declares no conflict of interest.

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
