# Peer review of "Optimal Period for Winter Mountain Tourism in Romania"

_sustainability, doi:10.3390/su14073878_

Round 1
Reviewer 1 Report
The undertaken research problem is important and interesting. The research results can be a source for comparative research in other countries and regions.
The text, however, is not fully scientific:
- the aim seems obvious, I suggest adopting specific aims or indicating an ex ante aim regarding research results,
- there is no hypothesis/hypotheses research.
- the methodology is very poorly presented,
- literary studies are very limited,
- the research results were not compared with other studies, also in other areas,
- therefore, there is no discussion, the part called Discussion is a presentation of research results,
- the Conclusion is too general and does not contribute much to science.
- In addition, editing of figures that are unreadable and/or not very communicative should be improved, this applies to: Figures 1, 5, 6 and Figure 2.
- In line 51, the period 2016-2018 is indicated as two years.
Author Response
Thank you for suggestions for improving this work. Following your answers, there were made the following changes.
- Was discussed the issue in the ex ante context, considering the event uncertain of climate change.
- Were presented the hypothesis of statistical research
- Were presented additional details in the research methodology. Furthermore, were inserted a figure describing this process.
- Was extended the study of literature with other consulted works.
- Were also presented other comparisons.
- Were added new items to the "Discussion" section. Some results were inserted incorrectly in the "Discussions" section and moved to "Results" section.
- Were extended and customized the conclusions.
- Were modified figures 1, 2, 5 and 6
- Were indicated the seasons 2016-2017 and 2017-2018 instead of two years.
Kind regards.
Reviewer 2 Report
Dear Editorial Board, Dear Authors,
The paper entitled “OPTIMAL PERIOD FOR WINTER MOUNTAIN TOURISM IN 2 ROMANIA“ aims to assess the extent to which climate change has changed the calendar period in which snow sports can be practiced in Romania.
The article addresses the important issue of climate change and the associated snow cover in Romania.
These changes cause the intensity of winter tourism to change. There is therefore a need to adapt the winter tourism and winter sports industry to vacations and school breaks.
The changes of the WINTER MOUNTAIN TOURISM period also concern other countries, e.g. Alpine countries in Europe and in the world
After reading the paper, I have comments and suggestions to improve the paper as follows:
In Introduction
- there is a lack of thorough introduction to the research problem based on world literature. No reference to other areas in the world with similar changes. This study is pioneering with respect to Romania. However, the journal “Sustainablity” has a global character and the research problem of the article should not only concern regional problems.
It would be appropriate to deepen the discussed research issue based on international literature, especially since the article does not have a separate chapter on Theoretical Background or Literature Review.
The paper does not ask detailed research questions.
In Materials and Methods
There is no figure that presents the research procedure.
All research methods used are not presented and described.
The author has shown very well by means of appropriate statistical methods to what extent climate change has changed the calendar period in which winter sports can be practiced in Romania.
The Results
Climate change were presented and described in a very good manner and are very interesting. They contribute to the value of this paper.
But winter tourism was presented only with one indicator (distribution of the number of overnight stays for tourist accommodation facilities). I propose to add information concerning the tourist offer, quality of services, type and category of accommodation facilities.
In the Discussion Section, the authors should discuss and explain the findings and results of the paper more. It also important to describe the results of the paper in greater detail in this section. This would contribute to a high improvement of this paper. The authors should compare their project and results with results from similar conducted research on this topic from other parts of Europa and all around the world.
What does this research bring to the tourism industry in Romania?
I propose to add recommendations and indicate possibilities and directions for change in winter tourism.
Technical errors that need to be removed:
[138] Figure 2 is illegible, it needs to be corrected
Kind regards,
Author Response
Thank you for suggestions for improving this work. Following your answers, there were made the following changes.
In Introduction
Were added data from the literature. Were assumed that the related results calculated for Romania's relief and geographical position could provide a model for other similar locations.
In Materials and Methods
Were presented additional details in the research methodology. Furthermore, were inserted a figure describing this process.
The Results
Were added new items to the "Results" section. Some of them were inserted incorrectly in the "Discussions" section.
In the Discussion Section
Were continued the series of discussions. Were also presented other comparisons.
In Conclusion section there were indicated other possibilities for improve in winter tourism.
Technical errors - figure 2 (current figure 3) has been replaced.
Kind regards.
Reviewer 3 Report
General Comments: The paper aims to understand the optimal period to set the winter break. The author use data from the National Meteorological Administration Romania to assess that optimal period. Results of the paper show that weather condition that are optimal for the winter sport season in Romania can be witnessed during January and the first part of February. According to the author, school and labor vacations’ must be adjusted then to cover those time periods instead of the traditional ones (that start late December and end around the second Thursday of January).
The general reason is that the motivation and importance of the research is not highlighted enough. The author claims that vacations’ schedules should be changed if the weather conditions that are optimal for winter sports are not present during the traditional schedule. However, possibly the traditional schedule exists not only to cover for the most suitable time for winter sports and optimal winter weather but also for other reasons. These other reasons might be that people want to have a vacation during Christmas and have one on New Year’s Eve. Since most of the Romanian population (around 80 percent if we consider world atlas a credible source) are Christians, assuming that that majority would prefer the traditional schedule is a valid assumption. In addition, changing the traditional schedule might change consumption during that period, without knowing if the change is for the best or worse. Thus, additional research is needed to understand what will maximize people’s welfare concerning vacation schedule to reach efficient policy implications.
Specific Comments
Abstract: Specify what is meant by “the first part of February”. Does it mean the first week, ten days, or half?
Introduction: The sentence “Climate change is directly affecting this type of tourism in multiple locations worldwide.” Citation is needed
Material and Methods: It might be more professional to write “Methodology” or “Method” instead of the current title.
- “Statistical calculations were performed using SAS Studio software”, state what are those statistical calculation, why are they important, and what is the goal.
- “In the Western Carpathians, the highest peak is Pietrosu, reaching a maximum height of 2303 metres. In the Western Carpathians, the highest peak is Bihor, 1849 metres.”. A clear mistake in defining the area can be seen in this sentence. The Eastern Carpathians’ highest peak is Petrosu (in the area labeled 3 in the figure one in the paper). The Western Carpathians highest peak is Cucurbăta Mare with an altitude of 1849 meters. Bihor is the name of the mountains that have ,Cucurbăta Mare as their highest peak. (The information can be found on mindat organization, google maps, and all commercial websites that appear with a google search).
- Further information on how the method was conducted is needed. Some Elaboration is found later in the results part that must be under methods. Especially describing how the models used transform the raw data into information. This must be carried to allow all reader from different background to understand well the methods. It will increase the credibility of the results.
Results: Describe what does the NSIR analyses data and what is it exactly so that all readers grasp the concept.
- The first figure in the results (figure 2) is messy. The numbers of both axes are mixed on top of each other, making harder to read. I would recommend a simple line chart instead.
- The paragraph under figure 2 should be in the methods not results. Example” the Kruskal-Wallis test was used, applying the proce-145 dure SAS Studio 3.8, Non Parametric One-Way ANOVA.”. The results must describe the output at hand and not the tools and models used to obtain it. Same comment applies to the following the figures.
- The sentence: “It should be noted that on December 1, only 266 about 19% of the surface of Romania is covered by negative temperatures. On December 267 24, when the winter holidays start, the estimated negative temperature coverage is about 268 45%.” Was already mentioned previously. Use terms as “As stated before” to restate the information if it is of high importance.
Conclusion: Missing limitations.
-It is stated that the solution is difficult to implement due to certain reasons. Suggest some possible policy implications that might help.
Language: Try using shorter sentence, it would be better for readers to follow up. Use per cent instead of the symbol %. Align the figures with the text (especially the first figure that seems out of line).
Author Response
Thank you for suggestions for improving this work. Following your answers, there were made the following changes.
In Abstract: The expression “the first part of February” has been removed. It refers to the months of January and February.
In Introduction: The source was indicated for the sentence “Climate change is directly affecting this type of tourism in multiple locations worldwide.”
In Material and Methods:
The change has been made in “Methodology”.
Were presented additional details in the research methodology.
We replaced the name of the mountain peak Curcubata Mare.
In Results:
Were added new items to the "Results" section.
Figure 2 (current figure 3) has been replaced.
The paragraph indicating the use of the Kruskal Wallis test has been moved to the Methodology section.
Was used the term "As stated before" in order to restate information.
In Conclusion:
Was added a comment on limitations. Suggestions have been given.
In Language:
Shorter sentences were used where possible. The % symbol has been replaced. The figures have been aligned with the text.
Kind regards.
Round 2
Reviewer 1 Report
Thank you to the Authors for their response to the review.
Reviewer 2 Report
My suggestions and comments have been incorporated into the article.
Kind regards
Reviewer 3 Report
I would like to thank the authors for taking into account my comments.